# Shape And Structure Preserving Differential Privacy

**Carlos Soto**
Department of Statistics
Pennsylvania State University
University Park, PA
cjs7363@psu.edu

**Karthik Bharath**
School of Mathematical Sciences
University of Nottingham
Nottingham, UK
Karthik.Bharath@nottingham.ac.uk

**Matthew Reimherr**
Department of Statistics
Pennsylvania State University
University Park, PA
mreimherr@psu.edu

**Aleksandra Slavkovic**
Department of Statistics
Pennsylvania State University
University Park, PA
sesa@psu.edu

## Abstract

It is common for data structures such as images and shapes of 2D objects to be represented as points on a manifold. The utility of a mechanism to produce sanitized differentially private estimates from such data is intimately linked to how compatible it is with the underlying structure and geometry of the space. In particular, as recently shown, utility of the Laplace mechanism on a positively curved manifold, such as Kendall's 2D shape space, is significantly influenced by the curvature. Focusing on the problem of sanitizing the Fréchet mean of a sample of points on a manifold, we exploit the characterisation of the mean as the minimizer of an objective function comprised of the sum of squared distances and develop a K-norm gradient mechanism on Riemannian manifolds that favors values that produce gradients close to the the zero of the objective function. For the case of positively curved manifolds, we describe how using the gradient of the squared distance function offers better control over sensitivity than the Laplace mechanism, and demonstrate this numerically on a dataset of shapes of corpus callosa. Further illustrations of the mechanism's utility on a sphere and the manifold of symmetric positive definite matrices are also presented.

## 1 Introduction

The amount of publicly available data has increased exponentially over the past decade and alongside with it the need for data privacy has emerged. As data gets increasingly more complex from scalar values to data on nonlinear manifolds, such as images, shapes and covariance matrices, there is a need for data privacy algorithms to adapt to the nonlinearity in, and preserve the geometric structure of, the data or parameter space. Intuitively, when the geometry of the manifold significantly influences sensitivity bounds through curvature-dependent terms, one would expect a structure-preserving privacy mechanism developed directly on manifolds to have better utility than its Euclidean counterpart on higher-dimensional ambient spaces within which the manifold is embedded. This was observed in recent work on a Laplace mechanism on manifolds [Reimherr et al., 2021], and especially for manifolds with positive curvature.

Apart from spherical, or directional, data, an archetypal example of data on a positively curved manifold arises in statistical shape analysis of planar configurations representing 2D objects; the Kendall shape space of 2D points [Kendall, 1984] modulo shape-preserving similarity transformations (e.g., rotations) is a Riemannian manifold with positive curvature. It is evident in this setting that

36th Conference on Neural Information Processing Systems (NeurIPS 2022).

the utility of a privacy mechanism will depend on how well it is compatible with shape preserving transformations of the data—sanitized versions of a shape summary of a 2D image of a bird should 'look' like a bird, impervious to (global) rotation, scaling and translation.

Structure-preserving mechanisms within specific contexts have been considered before; see, for example Jiang et al. [2016] concerning covariance matrices; Imtiaz and Sarwate [2018, 2016], Gilad-Bachrach and Gonen [2017], Awan et al. [2019], Chaudhuri et al. [2013], Biswas et al. [2020] for private principal component analysis; and Sheffet [2015] for private linear regression. A general structure-preserving Laplace mechanism on manifolds was considered by Reimherr et al. [2021], in Awan and Slavković [2021] the relationship between the data space and the sensitivity was examined in sufficient generality, and a general Gaussian mechanism on Riemmanian manifolds for approximate differential privacy was considered by Han et al. [2022]. However, research on privatizing shape summaries with theoretical guarantees is conspicuous in its absence within the privacy literature; the most relevant one we have found comes from computer vision which produces random faces but offers no differentially private guarantees [Karras et al., 2019].

The two-fold motivation for our paper is to a develop a privacy mechanism for statistical shape analysis, and more generally for data on manifolds, that is compatible with the geometry of the underlying space, and further, offers better control on how the curvature influences global sensitivity bounds. To this end, our main contributions are as follows.

1. We develop an extension of the K-norm Gradient Mechanism (KNG) on $\mathbb{R}^d$ [Reimherr and Awan, 2019] for producing sanitized private estimates under the pure differential privacy framework to the setting of Riemannian manifolds, with a focus on mean estimation.
2. We derive a curvature-dependent upper bound on global sensitivity, which for the important case of positively curved manifolds is smaller than the corresponding one arising from a recently proposed Laplace mechanism [Reimherr et al., 2021]. Our numerical examples verify that the KNG mechanism on manifolds shares the powerful utility of its counterpart on $\mathbb{R}^d$, and this is is tied to the curvature of the manifold and not to the dimension of the ambient space.
3. We introduce the first, to our knowledge, differentially private shape analysis under Kendall's 2D shape space framework, and favorably compare its performance to mechanisms designed for the higher-dimensional ambient space (and not directly on the manifold) on a dataset of corpus callosa obtained from MR images.

## 2 Background

In this section we introduce the necessary tools from differential geometry and differential privacy as well as the notation for this paper. For a thorough exposition of differential geometry and shape analysis we refer to Do Carmo [1992], Srivastava and Klassen [2016] and for DP we refer to Dwork and Roth [2014].

### 2.1 Differential geometry

Let $\mathcal{M}$ be a complete, connected Riemannian manifold of dimension $d$. Denote by $T_m\mathcal{M}$ the tangent space at each point $m \in \mathcal{M}$ and by $T\mathcal{M} = \{T_m\mathcal{M} : m \in \mathcal{M}\}$ the collection of tangent spaces, known as the tangent bundle. On the tangent space $T_m\mathcal{M}$ at each point $m$, we can define an inner product $\langle \cdot, \cdot \rangle_m : T_m\mathcal{M} \times T_m\mathcal{M} \to \mathbb{R}$ with induced norm $\| \cdot \|_m$; the collection $\{\langle \cdot, \cdot \rangle_m : m \in \mathcal{M}\}$ is referred to as a Riemannian metric. The Riemannian metric varies smoothly along the manifold and allows us to measure distances, volumes, and angles.

For a curve, or path, $\alpha : [0, 1] \to \mathcal{M}$, the vector $\alpha'(t)$ is its instantaneous velocity, and its length $L(\alpha)$ is the value $\int_0^1 \|\alpha'(t)\|_{\alpha(t)}^{1/2} \mathrm{d}t$. A curve $\alpha$ is said to be arc-length parameterised if $\|\alpha'(t)\|_{\alpha(t)} \equiv 1$ and thus $L(\alpha(0), \alpha(t_0)) = t_0$. Geodesic curves are those with zero acceleration for all $t$. The distance $\rho$ between two points $p$ and $q$ is the length of the shortest path, a segment of a geodesic curve connecting the two known as the minimal geodesic: $\rho(p, q) = \inf\{L(\alpha)|\alpha : [0, 1] \to \mathcal{M}; \alpha(0) = p, \alpha(1) = q\}$.

Given a point $p$ and a geodesic $\alpha$ with $\alpha(0) = p$, a cut point of $p$ is defined as the point $\alpha(t_0)$ such that $\alpha$ is a minimal geodesic on the interval $[0, t_0]$ but fails to be for $t > t_0$. The set of all cut points of geodesics starting at $p$ is its cut locus. The injectivity radius of $p$ is the distance to its cut locus, and the injectivity radius inj $\mathcal{M}$ of $\mathcal{M}$ is the infimum of the injectivity radii of all points in $\mathcal{M}$.

The next two tools are necessary for moving on the manifold, moving to and from the tangent spaces, and are particularly useful for sampling from distributions on manifolds. For a geodesic $\alpha$ starting at $p$ with initial velocity $v$, the *exponential map* $\exp(p, \cdot) : T_p\mathcal{M} \rightarrow \mathcal{M} :$ is defined as $\exp(p, v) = \alpha(1)$. From the Hopf-Rinow theorem, on a complete manifold the exponential map is surjective. On an open ball around the origin in $T_pM$ it is a diffeomorphism onto its image outside of the cut locus of $p$, and a well-defined inverse $\exp^{-1}(p, \cdot) : \mathcal{M} \rightarrow T_p\mathcal{M}$ exists, known as the *inverse exponential* or logarithm map, that maps a point on $\mathcal{M}$ outside of the cut locus of $p$ to $T_pM$; thus for any $q$ outside of the cut locus of $p$, $\rho(p, q) = \| \exp^{-1}(p, q) \|_p$.

There are many notions of curvature of a Riemannian manifold. We will mainly be concerned with *sectional curvature* at a point $p$, defined to be the Gaussian curvature at $p$ of the two-dimensional surface swept out by the set of all geodesics starting at $p$ with initial velocities lying in the two-dimensional subspace of $T_p\mathcal{M}$ spanned by two linear independent vectors.

Further, volumes of sets can be computed using the Riemannian volume form $\mathrm{d}\mu$. In local coordinates, the coordinate-independent Riemannian volume form is defined as $\sqrt{\det(g)}\mathrm{d}x_1 \wedge \mathrm{d}x_2 \cdots \wedge \mathrm{d}x_d$, where $g_{ij} = \langle \partial x_i, \partial x_j \rangle$ is the Riemannian metric tensor. A vector field on $\mathcal{M}$ is a differentiable mapping $\mathcal{M} \rightarrow T\mathcal{M}$ that assigns to each point $m$ on $\mathcal{M}$ a tangent vector in $T_m\mathcal{M}$. Suppose we have a smooth function $h$ defined over $\mathcal{M}$, the gradient $\nabla h$ of $h$ is the vector field defined by the relationship $\langle \nabla h(p), v \rangle_p = \partial h_p(v)$ for $p \in \mathcal{M}$ and $v \in T_p\mathcal{M}$.

## 2.2 Differential privacy

Let $D = \{x_1, \ldots, x_n\} \subset \mathcal{M}$ denote a dataset of size $n$. In several statistical and machine learning problems, one of the most popular tools for releasing a sanitized version of a minimizer

$$\hat{\theta} = \mathrm{argmax}_{x \in \mathcal{M}} U(x; D)$$

of a utility function $U$ over $\mathcal{M}$ is the exponential mechanism introduced by McSherry and Talwar [2007], based on a density

$$f(x; D) \propto \exp\left\{\sigma^{-1} U(x; D)\right\},$$

where the scale or rate parameter $\sigma$ is chosen to achieve a desired level of privacy and accounts for the sensitivity of $U$. If $\mathcal{M}$ is the Euclidean space, the density is with respect to the Lebesgue measure, and for finite or countable $\mathcal{M}$, it is with respect to the counting measure.

A modification of the exponential mechanism is the K-norm Gradient Mechanism (KNG) introduced by Reimherr and Awan [2019], which turns out to have better utility quite generally. The idea is that the maximizer of $U(x; D)$ is also the point at which its gradient is zero. On a Riemannian manifold $\mathcal{M}$, a KNG mechanism can be constructed using the gradient vector field $\nabla U(x; D)$, where the gradient is defined with respect to the Riemannian metric, with the (unnormalised) density

$$f(x; D) \propto \exp\{-\sigma^{-1}\|\nabla U(x; D)\|_x\}$$

defined with respect to the volume measure [Reimherr et al., 2021], where $\| \cdot \|_x$ refers to the norm with respect to Riemannian metric at $x$, not be confused with the subscript $k$ as in a $k$-norm. For general manifolds $\mathcal{M}$, conditions on the sectional curvatures are typically needed to ensure that the density is integrable with a finite normalizing constant. Under such conditions we can introduce a definition of differential privacy similar to that of Blum et al. [2005].

**Definition 1.** *For $\epsilon > 0$, a privacy mechanism satisfies $\epsilon$-differential privacy (pure differential privacy, $\epsilon$-DP) if for any pair of adjacent databases $D$ and $D'$, denoted $D \sim D'$, we have that*

$$\int_S f(x; D)d\mu \leq e^\epsilon \int_S f(x; D')d\mu$$

*for any measurable set $S \subset \mathcal{M}$.*

To determine the rate parameter for the KNG mechanism, one needs to quantify the robustness or sensitivity, of the norm of gradient vector field for adjacent databases.

**Theorem 1.** *If for all neighboring $D \sim D'$ and almost all $x$ we have*

$$\|\nabla U(x; D) - \nabla U(x; D')\|_x \leq \Delta,$$

*then one can take $\sigma = 2\Delta/\epsilon$ so that the KNG mechanism will be $\epsilon$-DP. Here $\Delta$ is referred to as the global sensitivity.*

The proof of Theorem 1 follows directly from an application of the triangle inequality. The global sensitivity $\Delta$ plays a crucial role in determining the behavior of KNG about the optimizer of $U$. Thus far, $U$ has been a generic utility function however consideration needs to be taken for cases when $U$ does not have a global optimizer.

## 3 Differentially private Fréchet mean estimation theory

Possibly the most fundamental summary statistic is the average or mean. In a Euclidean setting, the mean has a closed form expression, which for general manifolds is rarely the case. Generalizing the notion of the Euclidean mean, the Fréchet mean is defined as the minimizer of the variance functional

$$F(\cdot, D) : \mathcal{M} \to \mathbb{R}_+, \quad F(x; D) := \frac{1}{2n} \sum_{i=1}^{n} \rho(x_i, x)^2,$$

where $\rho$ is the Riemannian distance. In general, the minimizer may not exist, and may not be unique when it does. Study of conditions that ensure existence and uniqueness has a long history (see for e.g., Karcher [1977] and Afsari [2011]), and we thus take some necessary precautions outlined in Assumption 1.

**Assumption 1.** *The dataset $D \subset B_r(p_0)$, a geodesic ball centered at $p_0$ with finite radius $r$, with $r < \frac{1}{2} \min\{\text{inj } \mathcal{M}, \frac{\pi}{2} \kappa_{\max}^{-1/2}\}$ and $\kappa_{\max} > 0$ is an upper bound on the sectional curvatures of $\mathcal{M}$.*

For non-positively curved $\mathcal{M}$, $\kappa_{\max}^{-1/2}$ is interpreted as $+\infty$, and Assumption 1 states that the data $D$ must be bounded. That is, the data can lie in a ball of any arbitrary size as long as we know how large the ball is as this directly affects the global sensitivity. The weaker requirement $r < 1/2 \min\{\text{inj } \mathcal{M}, \pi \kappa_{\max}^{-1/2}\}$ suffices to ensure existence and uniqueness of the Fréchet mean. However, the stronger Assumption 1 is required to ensure that $(x, y) \mapsto \rho^2(x, y)$ is convex along geodesics (geodesically convex) within $B_r(p_0)$ Le [2001]; for example, $\rho^2$ is convex when restricted to a ball of radius smaller than $\pi/4$ on unit spheres since $\kappa_{\max} = 1$. For $\rho^2$ to be *strong* geodesically convex, an additional lower bound on sectional curvatures of $\mathcal{M}$ is required (see Lemma 1).

For mean estimation a natural utility function is $U(x; D) = -F(x; D)$. The KNG mechanism makes use of the gradient of $U(x; D)$ at $x$ and hence the (Riemannian) gradient of $-F(x; D)$, which in-turn is linked to the gradient of square-distance function $x \mapsto \rho(x_i, x)^2$ for fixed $x_i$. Under Assumption 1, each $x_i$ lies within the injectivity radius of $x$, and the gradient $\nabla \rho(x_i, x)^2 = -2 \exp^{-1}(x, x_i)$. Thus

$$\nabla F(x; D) = -\frac{1}{n} \sum_{i=1}^{n} \exp^{-1}(x, x_i).$$

The KNG mechanism then samples from the density (with respect to the volume measure)

$$f(x; D) \propto \exp \left\{ -\frac{1}{\sigma n} \left\| \sum_{i=1}^{n} \exp^{-1}(x, x_i) \right\|_x \right\}$$

to release a private statistical summary of the mean; note that when restricted to an open ball as per Assumption 1, the density $f$ has a finite normalising constant that depends on $\sigma$.

In Theorem 2 we provide a bound for the the global sensitivity of the KNG mechanism for mean estimation on Riemannian manifolds. The bound is curvature-dependent in the sense that it depends on the radius $r$ of the ball in which the data lies, the sample size $n$, and a function of $r$ and $\kappa_{\max}$. Specifically, the bound is equivalent to the bound obtained for Euclidean spaces for non-positively curved $\mathcal{M}$ but is inflated for positively curved $\mathcal{M}$.

**Theorem 2.** *Under Assumption 1 let $D = \{x_1, x_2, \ldots, x_n\}$ and $D' = \{x_1, x_2, \ldots, x_n'\}$ be adjacent datasets . Then*

$$\|\nabla U(x; D) - \nabla U(x; D')\|_x \leq \frac{2r(2 - h_{\max}(2r, \kappa_{\max}))}{n} \qquad where$$

$$h_{\max}(s, \kappa_{\max}) := \begin{cases} s\sqrt{\kappa_{\max}} \cot(s\sqrt{\kappa_{\max}}), & \kappa_{\max} > 0 ; \\ 1, & \kappa_{\max} \leq 0 . \end{cases} \qquad (1)$$

*Proof.* For two adjacent databases $D \sim D'$ that, without loss of generality, differ in the last element we have that

$$\left\|\nabla U(x; D) - \nabla U(x; D')\right\|_x = \left\|\nabla F(x; D) - \nabla F(x; D')\right\|_x = \frac{1}{n}\left\|\exp^{-1}(x, x_n) - \exp^{-1}(x, x_n')\right\|_x$$

$$\leq \frac{1}{n} 2r(2 - h_{\max}(2r, \kappa_{\max})),$$

based on Jacobi field estimates from Karcher [1977]; see also [Reimherr et al., 2021, Lemma 1]. $\square$

**Remark 1.** For the Laplace mechanism on $\mathcal{M}$ with density $f(x; D) \propto e^{-\frac{1}{\sigma}\rho(x,\eta)}$ for a fixed point $\eta$, whenever $r$ is chosen as per Assumption 1, the magnitude of inflation of global sensitivity due to curvature was shown to be $2r(2 - h_{\max}(2r, \kappa_{\max}))/(nh_{\max}(2r, \kappa_{\max}))$ in Reimherr et al. [2021]. The upper bound obtained by the KNG mechanism is thus strictly smaller for positively curved spaces since $0 < h_{\max}(2r, \kappa_{\max}) < 1$; in our experiments (see 4.1), this difference in sensitivity appears to result in better utility. Although the densities for the KNG and the Laplace are not the same, they are related to the norm of a vector in $T_x\mathcal{M}$: when $\eta = \bar{x}$, the Fréchet mean, since $\rho(x, \bar{x}) = \|\exp^{-1}(x, \bar{x})\|_x$, the Laplace density uses the norm of the Fréchet mean $\bar{x}$ when projected onto $T_x\mathcal{M}$; on the other hand, the KNG density uses the norm of the sample mean of data $\{x_i\}$ when projected onto $T_x\mathcal{M}$.

Our next theorem shows that the utility of KNG on manifolds, as measured by the intrinsic distance $\rho(\tilde{x}, \bar{x})$ matches the optimal rate of $O(d/n\epsilon)$ as in Euclidean space. For the theorem however we require the following Lemma, proof of which is available in the Supplemental materials.

**Lemma 1.** *In addition to Assumption 1, assume that there exists a $\kappa_{\min} \in \mathbb{R}$ that lower bounds the sectional curvatures of $\mathcal{M}$. Denote by $\bar{x}$ the Fréchet mean of $D$. For every $x \in B_r(p_0)$,*

$$h_{\max}(2r, \kappa_{\max})\rho(\bar{x}, x) \leq \|\nabla U(x, D)\|_x \leq h_{\min}(2r, \kappa_{\min})\rho(\bar{x}, x),$$

*where*

$$h_{\min}(s, \kappa_{\min}) := \begin{cases} s\sqrt{|\kappa_{\min}|}\coth(s\sqrt{|\kappa_{\min}|}), & \kappa_{\min} < 0 ; \\ 1, & \kappa_{\min} \geq 0 . \end{cases} \tag{2}$$

**Theorem 3.** *Assume the setting of Lemma 1. Let $\tilde{x}$ denote a draw from the KNG mechanism restricted to $B_r(p_0)$, and $\bar{x}$ be the unique global optimizer of $U(x; D)$. We have that*

$$E\left[\rho(\tilde{x}, \bar{x})^2\right] = O\left(\frac{d^2}{n^2\epsilon^2}\right).$$

*Proof.* Recall that under Assumption 1, $f(x; D) = C_\sigma^{-1}\exp\left\{-\sigma^{-1}\|U(x; D)\|_x\right\}$ for a finite normalizing constant $C_\sigma^{-1}$. Then

$$C_\sigma \geq \int_{B_r(p_0)} \exp\left\{-\sigma^{-1}h_{\min}(2r, \kappa_{\min})\rho(x, \bar{x})\right\} d\mu(x),$$

which due to Lemma 1 results in the upper bound

$$E\left[\rho(\tilde{x}, \bar{x})^2\right] \leq \frac{\int_{B_r(p_0)} \rho(x, \bar{x})^2 e^{-\frac{1}{\sigma}h_{\max}(2r, \kappa_{\max})\rho(x, \bar{x})} d\mu(x)}{\int_{B_r(p_0)} e^{-\frac{1}{\sigma}h_{\min}(2r, \kappa_{\min})\rho(x, \bar{x})} d\mu(x)}. \tag{3}$$

Since $r \leq \frac{1}{2}\mathrm{inj}M$, there exists a unique $v_x \in \exp^{-1}(\bar{x}, (B_r(p_0)) \cap \boldsymbol{B}_{2r}(\bar{x})$ such that $v_x = \exp^{-1}(\bar{x}, x)$, where $\boldsymbol{B}_{2r}(\bar{x}) = \{v \in T_{\bar{x}}M : \|v\|_{\bar{x}} < 2r\}$ is the open ball of radius $2r$ centred at the origin in $T_{\bar{x}}M$. Denote by $S_{\bar{x}}$ the bounded subset $\exp^{-1}(\bar{x}, (B_r(p_0)) \cap \boldsymbol{B}_{2r}(\bar{x})$ of $T_{\bar{x}}M$ with compact closure. With respect to the pushforward of $d\mu$ on to $T_{\bar{x}}M$ under the inverse exponential map at $\bar{x}$, the ratio in (3) equals

$$\frac{\int_{S_{\bar{x}}} \|v_x\|_{\bar{x}}^2 e^{-\frac{1}{\sigma}h_{\max}(2r, \kappa_{\max})\|v_x\|_{\bar{x}}} d(\mu \circ \exp(\bar{x}, \cdot))(v_x)}{\int_{S_{\bar{x}}} e^{-\frac{1}{\sigma}h_{\min}(2r, \kappa_{\min})\|v_x\|_{\bar{x}}} d(\mu \circ \exp(\bar{x}, \cdot))(v_x)}.$$

The induced measure $d(\mu \circ \exp(\bar{x}, \cdot))$ on $T_{\bar{x}}M$ can be extended to $\mathbb{R}^d$ by settings its value on the complement of $S_{\bar{x}}$ to be zero. It is then absolutely continuous with respect to the Lebesgue measure

$d\lambda$ on $\mathbb{R}^d$ with a Jacobian determinant uniformly bounded above and below, respectively, by constants $c_1$ and $c_2$ on (the closure) of $S_{\bar{x}}$. This ensures that the above ratio is upper bounded by

$$\frac{c_1}{c_2} \frac{\int_{\mathbb{R}^d} \|v_x\|_{\bar{x}}^2 e^{-\frac{1}{\sigma} h_{\max}(2r, \kappa_{\max})\|v_x\|_{\bar{x}}} d\lambda(v_x)}{\int_{\mathbb{R}^d} e^{-\frac{1}{\sigma} h_{\min}(2r, \kappa_{\min})\|v_x\|_{\bar{x}}} d\lambda(v_x)},$$

which, with a change of variables and using spherical coordinates, equals

$$\sigma^2 \frac{c_1}{c_2} \frac{h_{\min}(2r, \kappa_{\min})}{h_{\max}(2r, \kappa_{\max})^3} \left[ \int_0^\infty z^{d-1} e^{-z} d\lambda(z) \right]^{-1} \int_0^\infty z^{d+1} e^{-z} d\lambda(z) = O\left( \frac{d^2}{n^2 \epsilon^2} \right),$$

since the curvature-dependent terms $h_{\min}(2r, \kappa_{\min})$ and $h_{\max}(2r, \kappa_{\max})$, defined in (1) and (2), are both positive and finite under Assumption 1, $\sigma$ is as in Theorems 1 and 2, and the integrals in the numerator and denominator equal $(d+1)!$ and $(d-1)!$, respectively. $\qquad\square$

## 4 Examples

In this Section, we consider two simulated examples and a real data example on 2D shapes. For the former, we consider the positively curved unit $d$-sphere and the set of symmetric positive definite matrices (SPDM), which when equipped with an affine invariant metric is negatively curved. Manifold valued data naturally emerges in statistical applications such as with SPDM matrices in computer vision Caseiro et al. [2012] and brain diffusion tensor data Niethammer et al. [2006], discrete distributions can be modeled on a sphere, in network analysis Ginestet et al. [2017] for instance, and we refer to Younes [2012], Kimmel [2003] as references with a more complete overview.

In each scenario we sample from KNG via Metropolis-Hastings, a Monte Carlo Markov Chain method. This sampling method estimates our target distribution KNG and can incur a privacy loss. That is, KNG is an instantiation of the exponential mechanism and sampling from such a mechanism via MCMC produces approximate DP samples rather than pure DP samples Shen and Yu [2013], Wang et al. [2015]. The size of this cost is a function of the chain length, however, and decreases to zero at a geometric rate, so sampling must be done in a careful manner. An exact sampler would be ideal, however even for some cases of well-behaved distributions sampling is not so simple Seeman et al. [2021]. This issue is a limitation of this work and we leave this as an open problem.

### 4.1 Spheres

Let $\mathcal{S}_\kappa^d$ denote the $d$-dimensional sphere with radius $\kappa^{-1/2}$. The sphere equipped with the induced metric from $\mathbb{R}^{d+1}$, the canonical metric, has constant positive curvature $\kappa$. The tangent space at $p \in \mathcal{S}_\kappa^d$ is then $T_p \mathcal{S}_\kappa^d = \{v \in \mathbb{R}^{d+1} | \langle v, p \rangle = 0\}$. The exponential map, defined on all of $T_p \mathcal{S}_\kappa^d$, is given by $\exp(p, v) = \cos(\|v\|)p + \kappa^{-1/2}\sin(\|v\|)v/\|v\|$ and $\exp(p, \mathbf{0}) = p$. The set of points at a distance at least $\pi$ from $p$ constitutes its cut locus, which then is the singleton $\{-p\}$. With $\theta := \rho(p, q) = \cos^{-1}(\langle p, q \rangle)$, the inverse exponential map $\exp^{-1}(p, q) = \theta(\sin(\theta))^{-1}(q - \cos(\theta)p)$ at $p$ is hence defined only within the open ball around $p$ with radius $\pi/(2\kappa^{1/2})$.

Next, we consider the utility of KNG on a manifold and compare it to other sanitization techniques. We generate random samples $D$ from $S_1^2$ and compute the Fréchet mean $\bar{x}$, both as described in the Supplemental material. We set $\epsilon = 1$ and sanitize $\bar{x}$ with three separate methods; first with the proposed method KNG on manifolds to produce $\tilde{x}_{KNG}$, second with the Laplace on manifolds as in Reimherr et al. [2021] to produce $\tilde{x}_L$, and lastly embedding $\bar{x}$ into $\mathbb{R}^3$ and privatizing with the Euclidean Laplace to produce $\tilde{x}_E$. The latter almost surely will not be on the sphere, however, since the privacy guarantees are invariant to post-processing, we project the estimate back onto the sphere by normalizing as $\tilde{x}_E \to \tilde{x}_E/\|\tilde{x}_E\|$.

We compute several such replicates at different sample sizes and display the utility comparison in the first (left) panel of Figure 1, where the utility is measured using the average Euclidean distance $\|\bar{x} - \tilde{x}\|$ such that $\tilde{x}$ is a the respective sanitized estimate. We see that adding noise in the ambient space with the Euclidean Laplace adds the most noise, which may be attributed to the need to sanitize over an extra dimension. After post-processing $\tilde{x}_E$ by projecting onto the sphere, the Euclidean Laplace mechanism does have better utility than its manifold counterpart, but this is not unexpected since the sensitivity of the manifold Laplace for positively curved manifolds is inflated compared to

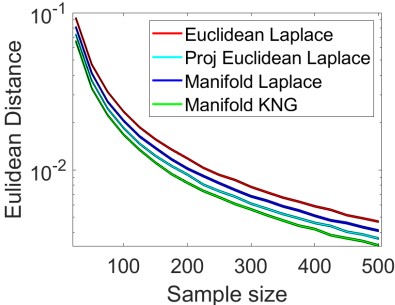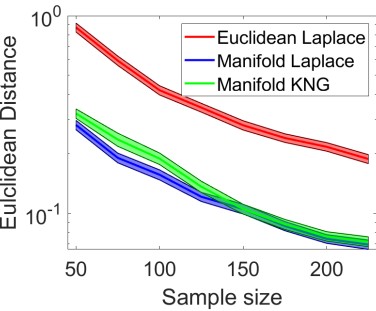

Figure 1: Utility measured using average Euclidean distance between the Fréchet mean $\bar{x}$ and its sanitized version $\tilde{x}$, when $\mathcal{M}$ is the unit sphere $\mathcal{S}_1^2$ in two dimensions (left) and $k \times k$ SPD matrices $\mathbb{P}(k)$ (right), under the following frameworks: (i) Manifold KNG; (ii) Euclidean Laplace by embedding $\bar{x}$ into the ambient space; (iii) Manifold Laplace on the manifold; and additionally, with (iv) Projected Euclidean Laplace for the unit sphere. For the Euclidean Laplace $\bar{x}$ was embedded into $\mathbb{R}^3$ for $\mathcal{S}_1^2$ and within $k \times k$ symmetric matrices for $\mathbb{P}(k)$. For each sample size, 10000 replicates were used for $\mathcal{S}_1^2$, whereas 500 were used for $\mathbb{P}(k)$. Shaded regions represent the average distance $\pm 2\text{SE}$, where the Euclidean distance for $\mathbb{P}(k)$ is $\|\text{vech}(\bar{x}) - \text{vech}(\tilde{x})\|$.

the Euclidean rate of $2r/n$ Reimherr et al. [2021]. Lastly, our proposed mechanism has the best utility in this comparison which may be attributed to its sensitivity being strictly less than the sensitivity of the Laplace on the manifold for positively curved manifolds. Further, our approach will always produce a private summary which is on the manifold and does not require any post-processing.

## 4.2 Symmetric positive-definite matrices

Denote by $\mathbb{P}(k)$ the $k(k+1)/2-$dimensional manifold of $k \times k$ symmetric positive-definite matrices equipped with the affine-invariant Rao-Fisher metric $\langle v, u \rangle_p = \text{Tr}(p^{-1}up^{-1}v)$, where $u, v \in T_p\mathbb{P}(k) = Sym_k$ are symmetric matrices. With this Riemannian metric $\mathbb{P}(k)$ has negative sectional curvature everywhere with exponential map $\exp(p, v) = p^{1/2}\text{Exp}\left(p^{-1/2}vp^{-1/2}\right)p^{1/2}$ and globally defined inverse exponential map $\exp^{-1}(q, p) = q^{1/2}\text{Log}\left(q^{-1/2}pq^{-1/2}\right)q^{1/2}$, where $\text{Exp}(\cdot)$ and $\text{Log}(\cdot)$ are the matrix exponential and logarithm, respectively. The distance between $q$ and $p$ in $\mathbb{P}(k)$ is thus $\rho(q, p) = \|\exp^{-1}(q, p)\|_q = \text{Tr}[\text{Log}(q^{-1/2}pq^{-1/2})^2]^{1/2}$.

For the simulations we set $k = 2$. We generate random samples $D \subset \mathbb{P}(k)$ by sampling from the Wishart distribution as discussed in the Supplemental materials. We compute the Fréchet mean $\bar{x}$ and sanitize it with three separate approaches: (i) we generate a private mean $\tilde{x}_{KNG}$ by sanitizing on $\mathbb{P}(k)$ using the proposed approach KNG, (ii) we generate a private mean $\tilde{x}_L$ on $\mathbb{P}(k)$ with the Laplace distribution as in Reimherr et al. [2021], Hajri et al. [2016], and (iii) we embed $\bar{x}$ into $Sym_k$ the space of symmetric matrices, represent $\bar{x}$ as a vector $\text{vech}(\bar{x}) \in \mathbb{R}^3$, sanitize by sampling from the Euclidean Laplace to produce $\text{vech}(\tilde{x}_E)$, and lastly we revert the vectorization to obtain $\tilde{x}_E$. There is no guarantee that $\tilde{x}_E$ will remain in $\mathbb{P}(k)$ and further without a unique projection to $\mathbb{P}(k)$ since it is an open cone within $\mathbb{R}^{k(k+1)/2}$.

Similarly to Section 4.1, in the second panel of Figure 1 we display an average utility comparison of the privatization techniques over 500 replicates with respect to the distance $\|\text{vech}(\bar{x}) - \text{vech}(\tilde{x})\|$, where $\tilde{x}$ is the sanitized estimate corresponding to one of the three approaches. We see that our approach has better utility compared to the Euclidean approach and comparable utility to the Laplace on $\mathbb{P}(k)$. The latter is not entirely surprising since the Laplace is equivalent to KNG for mean estimation in Euclidean space [Reimherr and Awan, 2019]. Sampling from the Laplace on $\mathbb{P}(k)$ is fairly simple since it was thoroughly studied by Hajri et al. [2016] and has nearly a closed form sampler; sampling from KNG on $\mathbb{P}(k)$ however is not as straightforward and we employed an Metropolis-Hastings algorithm which may account for its inconsistent behavior compared to the Laplace. However, our proposed method has better utility than the Euclidean approach, which is designed on the higher-dimensional ambient space.

## 4.3 Kendall's 2D shape space

Statistical shape analysis is a relatively recent field dating back to the seminal paper by Thompson and Thompson [1942] where shapes of animals, such as fish, were shown to differ in geometric transformations such as a shear. Since this conception there have been many branches of shape analysis that have arisen such as Kendall's shape space [Kendall, 1984], large deformation diffeomorphic metric mapping (LDDMM) [Grenander and Miller, 2007], and elastic shape analysis [Srivastava and Klassen, 2016]. No matter the choice, shape analysis has demonstrated to be widely applicable in the medical field (ERCAN et al. [2012], Li et al. [2014]), computer vision [Jimenez et al., 2000, Sharon and Mumford, 2006], and functional data analysis (Harris et al. [2021], Zhang and Srivastava [2020]). By "shape" of an object in two dimensions, we refer to the intrinsic geometric property of a set of points on the plane (representing the object) that remains unchanged under similarity transformations such as translation, rotation and scale Kendall [1984], and additionally on reparameterisations if an outline curve representation is used Srivastava and Klassen [2016]. Of the many areas of shape analysis available (for e.g., the theory of deformable templates [Trouvé and Younes, 2005] and Large deformation diffeomorphic metric mapping [Grenander and Miller, 2007]) we consider the Kendall shape space of two-dimensional landmark configurations Kendall [1984].

Consider a set $x = \{x_j\} \in \mathbb{C}^k$ of labelled $k$ points on a 2D object, known as landmarks, in the complex plane. If the object has been extracted from a densely sampled outline curve (for e.g., when extracted/segmented from a 2D image), then the parameterisation of the curve induces the labelling through an ordering of the points. Labelling thus establishes a correspondence between points on different objects, and is considered to be fixed.

The shape of $x$ is what remains once translation, scaling and rotation variabilities are removed or accounted for. Translation is removed by by transforming $x \to x - \frac{1}{k}\sum_{j=1}^k x_j$ resulting in a reduced space of the complex $(k-1)-$dimensional hyperplane $\mathcal{C} = \{x \in \mathbb{C}^k \backslash 0 | \frac{1}{k}\sum x_j = 0\}$. Scaling and rotation of $x$ amounts to multiplying by a complex number $re^{i\theta}$ where $r$ is the scaling factor and $\theta$ is the angle of rotation. The shape of $x$ then can be considered as the curve $(-\pi, \pi] \ni \theta \mapsto e^{i\theta}u$ on the complex unit $(k-1)$-dimensional sphere $\mathbb{C}S^{k-1}$ in $\mathcal{C}$, where $u = x/\|x\| \in \mathcal{C}$ (or the real sphere of dimension $2k-3$), $\|x\| = \sqrt{x^*x}$ and $x^*$ the complex conjugate of $x$. The shape space is thus identified with the compact complex projective space $\mathbb{C}P^{k-2}$ of dimension $k-2$ following the scaling $x \to x/\|x\|, x \in \mathcal{C}$. Therefore, the geodesic shape distance between landmark configurations $x$ and $y$ with corresponding centred and scaled versions $p$ and $q$ is $\rho(x,y) = \inf_{\theta \in (\pi, \pi]} \cos^{-1}\left(|e^{-i\theta}pq^*|\right)$; thus, the injectivity radius of the shape space is $\pi/2$.

Let $x$ be a centred and scaled configuration. Minimizing unit-speed geodesics starting at $x$ and initial velocity $v$ in the shape space are isometrically identified with unit-speed geodesics $(-\pi/2, \pi/2] \ni s \mapsto x\cos(s) + v\cos\sin(s)$ on $\mathbb{C}S^{k-1}$ wherein $v$ satisfies $v\mathbf{1}_k = 0$ in addition to $xv^* = 0$, and $\mathbf{1}_k$ is the vector of ones[Kendall et al., 1999, Chapter 6]; such geodesics are known as 'horizontal' geodesics. Consequently, the exponential map on the shape space is given by the corresponding one on the sphere with the additional condition on the velocity vectors. The inverse exponential map at $x$ exists within a ball of radius smaller than $\pi/2$ in the shape space, and is thus given by $\exp^{-1}(x,y) = \theta\|y - \mathrm{Proj}_x(y)\|$, where $\mathrm{Proj}_x(y) := x(y^*x)$ is the projection of $y$ onto $x$ and $\theta = \cos^{-1}|x^*y|$. The (complex) holomorphic sectional curvature of the 2D Kendall's shape space is constant and equals 4 [Kendall et al., 1999].

As an application we consider the pre-processed corpus callosum data of Cornea et al. [2017] from the Alzheimer's Disease Neuroimaging Initiative (ADNI). The data are from the mid-sagittal slices of MRIs (magnetic resonance images) and we refer to Cornea et al. [2017] on details of how the data was processed. Their data contains 409 total corpus callosa. The left portion of Figure 2 displays 10 sample corpus callosa where the parameterization is visually displayed as a color gradient from blue to yellow. In the middle we display the Fréchet mean of all 409 corpus callosa. Having computed the mean, we then sanitize the mean with three techniques: (i) using the proposed KNG mechanism (Right: top row); (ii) for a comparison with (i), sanitize each landmark of the mean using the Laplace mechanism splitting the privacy budget (Right: middle row); (iii) sanitize each landmark as in (ii) without factoring in rotational alignment in the corpus callosum. We expand on (ii) and (iii) in the Supplemental material.

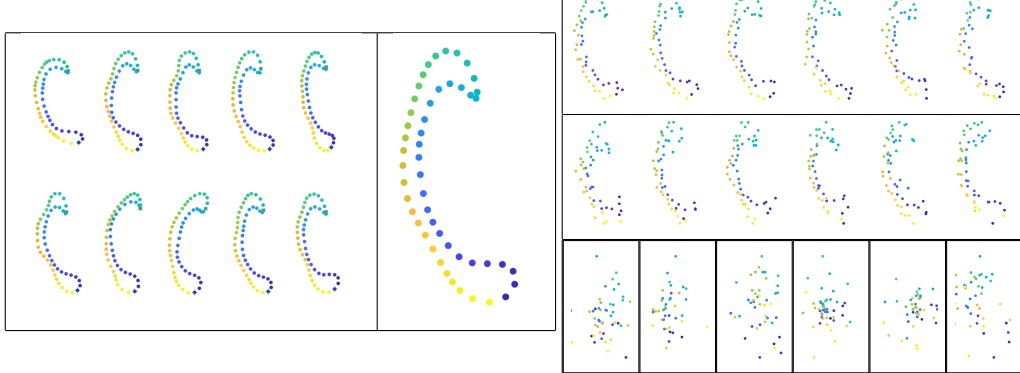

Figure 2: Left panel: A sample of ten corpus callosa from the sample of 409. Middle: The Fréchet mean corpus callosum. Right, top row: Six sample private corpus callosa privatized under the KNG framework on Kendall's 2D shape space. Right, middle row: Six sample private corpus callosa privatized point-wise with the Laplace mechanism. Right, bottom row: Six sample private corpus callosa privatized point-wise with the Laplace without accounting for rotational alignment. For more details, please refer to 4.3.

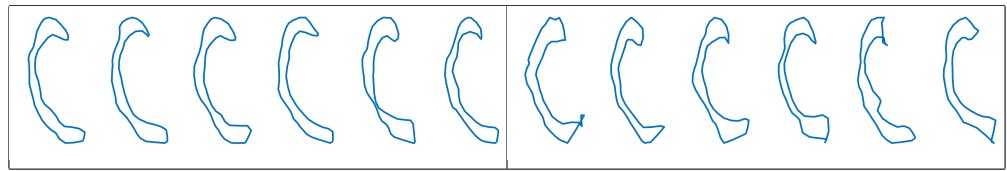

Figure 3: Differentially private corpus callosa estimates post-processed by using a first order local linear regression for smoothness. The left six are private under the proposed KNG framework while the right six are private under the point-wise Laplace as explained in Section 4.3.

As we can observe in Figure 2, when one does not account for shape-preserving transformations (rotations alignment in this example), the shape of the corpus callosum is entirely destroyed during sanitization. Sanitization over the shape manifold (Right: top row) tends to retain the structure of corpus callosum when compared with sanitizing over each coordinate (Right: middle row), which appears more distorted; for example, we observe that the shape of the fifth corpus callosum (Right: middle row) has a contour with crossings. Further, we post-process to smooth the private estimates by a first order local linear regression. Left panel of Figure 3 displays the post-processed KNG private estimates while the right panel shows point-wise Laplace private estimates, where rotational alignment has been carried out. We do not post-process the private estimates which do not consider rotational alignment as they appear to be non informative. Considering the mean shape in Figure 2, the "hook" at the top is quite prominent and comparing this to the private corpus callosa of Figure 3 we notice that the KNG estimates tend to preserve this structure. Even though all the private estimates are processed in the same way, we notice the Laplace estimates are not only less smooth but can possess undesirable distortions, such as additional features such as loops.

## 5 Conclusions and future work

In this paper we demonstrate that versatility and powerful utility of the K-norm gradient mechanism on $\mathbb{R}^d$ carries over to the manifold setting. In particular, better control over global sensitivity when compared to the recently introduced manifold Laplace mechanism [Reimherr et al., 2021] for positively curved manifolds motivates the development of, to our knowledge, the first privacy mechanism for statistical shape analysis of 2D point configurations. Gains in utility when working directly on the manifold, as opposed to the higher-dimensional ambient space, are observed in the numerical examples: in terms of utility, the KNG not only outperforms the Euclidean mechanism but also the manifold Laplace mechanism. Further, the Laplace on the manifold and our mechanism are intricately connected as they both are the exponential of a norm in a particular tangent space as we

note in Remark 1; this similarity in formulation and difference in sensitivity is tied to the better utility in the case of positively curved manifolds.

Depending on the manifold, statistical utility gains enjoyed by working on the manifold can be tempered by expensive geometric computation. For example, in the case of SPDM, the clear gains in utility are obtained within the context of computationally expensive sampling from the KNG (see also Supplemental material) owing to repeated computations of matrix inverses and square roots related to the exponential, inverse-exponential maps and the geodesic distance. Indeed in practice, however, only a single instantiation suffices. In contrast, sampling from the Laplace on manifolds is straightforward, and there is thus room for improvement in sampling from the KNG on manifolds.

Our work represents a step in the right direction in developing privacy mechanisms for a myriad of approaches to state-of-the-art statistical shape analysis on infinite-dimensional manifolds of curves and surfaces [Srivastava and Klassen, 2016], and diffeomorphisms [Grenander and Miller, 2007]. Moreover, our work opens up the possibility of developing geometry-driven privacy mechanisms for standard data analytic procedures used in various applications, such as principal component analysis (Grassmannian manifold of subspaces), rank-constrained matrix completion (quotient manifold of nonsingular matrices), and optimizing the Rayleigh quotient (Grassmannian), Procrustes problem (manifold of orthogonal matrices or frames), and pose estimation in computer vision (manifold of rotation matrices).

## Acknowledgments and Disclosure of Funding

This work was funded in part by NSF SES-1853209 (to CS, MR, AS), NSF DMS-2015374 and EPSRC EP/V048104/1 (to KB).

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
