# Supplemental to *Shape and Structure Preserving Differential Privacy*

## 1 Proof of Lemma 1

Recall that $f : \mathcal{M} \to \mathbb{R}$ is strong geodesically convex with parameter $\lambda > 0$ if for every $x, y \in \mathcal{M}$

$$f(y) \geq f(x) + \left\langle \nabla f(x), \exp^{-1}(x,y) \right\rangle_x + \frac{\lambda}{2} \rho^2(x,y);$$

the above definition is interpreted in a local sense within a suitable neighbourhood in which the inverse exponential map is well-defined. We first establish that $x \mapsto U(x, D)$ is strong geodesically convex and derive upper and lower bounds on its Hessian within $B_r(p_0)$ with $r$ chosen as per Assumption 1.

Note that when $s < \frac{\pi}{2\sqrt{\kappa_{\max}}}$, the function $s \mapsto h_{\max}(s, \kappa_{\max}) > 0$, decreasing, and bounded above by 1, while $s \mapsto h_{\min}(s, \kappa_{\min})$ is bounded below by 1 and increasing. Thus $0 < h_{\max}(2r, \kappa_{\max}) \leq h_{\max}(\rho(x,y), \kappa_{\max})$ for every $x, y \in B_r(p_0)$ under the assumption on $r$. On the other hand, $h_{\min}(2r, \kappa_{\min}) \geq h_{\min}(\rho(x,y), \kappa_{\min}) \geq 1$ for every $x, y \in B_r(m_0)$. For each $x_i \in B_{p_0}(r)$, from Lemma 1 of [1] we have that for $x, y \in B_r(m_0)$,

$$\rho^2(x, x_i) \geq \rho^2(y, x_i) + \langle \nabla \rho^2(y, x_i), \exp^{-1}(y, x) \rangle_y + \frac{2 h_{\max}(\rho(x, x_i), \kappa_{\max})}{2} \| \exp^{-1}(y, x) \|_y^2$$

$$\geq \rho^2(y, x_i) + \langle \nabla \rho^2(y, x_i), \exp^{-1}(y, x) \rangle_y + \frac{2 h_{\max}(2r, \kappa_{\max})}{2} \| \exp^{-1}(y, x) \|_y^2. \quad (1)$$

Summing over $i$ and dividing by $2n$ we get

$$U(x, D) \geq U(y, D) + \langle \nabla U(y, D), \exp^{-1}(y, x) \rangle_y + \frac{h_{\max}(2r, \kappa_{\max})}{2} \| \exp^{-1}(y, x) \|_y^2,$$

which implies that $U$ is strong geodesically convex with parameter $h_{\max}(2r, \kappa_{\max})$ inside $B_r(p_0)$. Thus, in local coordinates

$$\nabla^2 U(x, D) \succcurlyeq h_{\max}(2r, \kappa_{\max}) \mathbb{I}_d.$$

From assumption on the lower bound $\kappa_{\min}$ on sectional curvatures ensures, using Lemma 5 of [3] derived for non-positively curved manifolds, we obtain

$$U(x, D) \leq U(y, D) + \langle \nabla U(y, D), \exp^{-1}(y, x) \rangle_y + \frac{h_{\min}(2r, \kappa_{\min})}{2} \| \exp^{-1}(y, x) \|_y^2.$$

In other words, $U$ has a gradient that is geodesically Lipschitz with parameter $h_{\min}(2r, \kappa_{\min})$. As a consequence,

$$\nabla^2 U(x, D) \preccurlyeq h_{\min}(2r, \kappa_{\min}) \mathbb{I}_d.$$

On flat manifolds (e.g., $\mathbb{R}^d$, flat torus, cylinder) where $\kappa_{min} = \kappa_{\max} = 0$ we have $h_{\min}(2r,0) = h_{\max}(2r,0) = 1$, and $U$ is strong gedesically convex with parameter 1 when $D$ is restricted to lie within a ball of any finite radius. Summarily, in local coordinates,

$$h_{\max}(2r,\kappa_{\max})\mathbb{I}_d \preccurlyeq \nabla^2 U(x,D) \preccurlyeq h_{\min}(2r,\kappa_{\min})\mathbb{I}_d, \quad \forall x \in B_r(m_0). \tag{2}$$

We now consider the norm of the gradient vector field $\nabla U$. Let $\gamma$ be a unit-speed geodesic from $\bar{x}$ to $x$, and denote by $\Gamma_{\bar{x}}^x : T_{\bar{x}}M \to T_x M$ the parallel transport along $\gamma$; from our assumption on the radius $r$, the geodesic lies entirely within $B_r(p_0)$. Then,

$$\begin{aligned}
\|\nabla U(x,D)\|_x &= \big|\|\nabla U(x,D)\|_x - \|\Gamma_{\bar{x}}^x \nabla U(\bar{x},D)\|_x\big| \\
&\leq \|\nabla U(x,D) - \Gamma_{\bar{x}}^x \nabla U(\bar{x},D)\|_x \\
&\leq h_{\min}(2r,\kappa_{\min})\rho(\bar{x},x) \,.
\end{aligned}$$

The equality is due to the fact that the parallel transport map is an isometry between tangent spaces and fixes the origin; the first inequality follows from the reverse triangle inequality, while the last follows from the upper bound on the Hessian of $U$ in (2).

To derive the lower bound on $\|\nabla U(x,D)\|_x$, note that under our assumption on the radius $r$, the function $U$ is strong geodesically convex within $B_r(p_0)$ since the function $h_{\max}$ is positive. From the lower bound on the Hessian of $U$ in (2), for any $y \in B_r(p_0)$, we hence obtain

$$\left|\big\langle \nabla U(x,D) - \Gamma_y^x \nabla U(y,D),\ \exp^{-1}(x,y)\big\rangle_x\right| \geq h_{\max}(2r,\kappa_{\max})\rho(x,y)^2,$$

where $\Gamma_y^x$ is the parallel transport along a geodesic from $y$ to $x$; applying Cauchy-Schwarz to the inner product results in

$$\|\nabla U(x,D) - \Gamma_y^x \nabla U(y,D)\|_x \rho(x,y) \geq h_{\max}(2r,\kappa_{\max})\rho(x,y)^2.$$

Dividing both sides by $\rho(x,y)$ and taking $y = \bar{x}$ leads to the desired lower bound, since, within the local coordinates at $x$, $\Gamma_{\bar{x}}^x \nabla U(\bar{x},D)$ is the zero gradient vector field under the isometric parallel transport.

## 2 Simulation details

Simulations pertaining to the sphere and Kendall shape space are done on a desktop computer with an Intel Xeon processor at 3.60GHz with 31.9 GB of RAM running windows 10. Simulations pertaining to symmetric positive-definite matrices were performed on the Pennsylvania State University's Institute for Computational and Data Sciences' Roar supercomputer. All simulations are done in Matlab. This content is solely the responsibility of the authors and does not necessarily represent the views of the Institute for Computational and Data Sciences.

### 2.1 Fréchet Mean

To compute the Fréchet mean, we use the standard gradient descent approach. Given a sample $D = \{x_1, x_2, \ldots, x_n\}$ we initialize $\hat{\mu}_1$ at a data point. The variance functional which we wish to minimize is $U(x; D) = -\frac{1}{2n}\sum_{i=1}^n \rho^2(x,x_i)$. So, at iteration $k$ one takes a step in the direction of $t_k \nabla U(\hat{\mu}_{k-1}; D)$ where $t_k \in (0,1]$ from $\mu_{k-1}$. That is, $\hat{\mu}_k =$

$\exp(\hat{\mu}_{k-1}, t_k \nabla U(\hat{\mu}_{k-1}; D))$. To check convergence one could either check if the distance between adjacent iterations is smaller than some value $\beta_1 > 0$ or if the norm of the gradient is smaller than some value $\beta_2 > 0$, that is if $\rho(\hat{\mu}_1, \hat{\mu}_2) < \beta_1$ or if $\|\nabla U(\hat{\mu}_k; D)\|_{\hat{\mu}_k} < \beta_2$. The latter is convenient from a computational standpoint and how we measure convergence. We set $\beta_2 = 10^{-5}$ and $t_k = 0.5$ for all $k$. Further, to avoid a computational timeout we set a maximum number of iterations to 500 however for all examples the algorithm converged within the first couple hundred iterations. Lastly, we assume the data follows Assumption ?? and thus convergence issues and local minima pose no observed issues.

## 2.2 The Euclidean Laplace

A standard distribution to generate differentially private estimates is the Euclidean Laplace which is a K-norm mechanism with the $\ell_2$ norm. We sample from the distribution $f(x) \propto \exp\{-\sigma^{-1}|x - \bar{x}|\}$ on $\mathbb{R}^d$ as in [2].

1. Sample a direction $V$ uniformly from $\mathcal{S}^{d-1}$.

2. Sample a radial length $r$ from $\Gamma(d, 1)$, the Gamma distrbution with $\alpha = d$ and $\beta = 1$.

3. Set $Y = \bar{x} + r\sigma V$.

$Y$ will then be a draw from $f(x)$, the $d-$dimensional Euclidean Laplace distribution. Note that $x/|x|$ is uniform on $\mathcal{S}^{d-1}$ when $x \sim N_d(\mathbf{0}_d, \mathbb{I}_d)$.

## 2.3 SPDM simulations

### 2.3.1 Generating random samples

We have that $\mathbb{P}(k)$ is the space of symmetric positive-definite matrices. We note that the Wishart distribution has support on $\mathbb{P}(k)$, and draw from $Y \sim W(V, df)$ where $E(Y) = V \cdot df$, $df > 0$, and $V$ is a symmetric $k \times k$ matrix. We require that the $D \subset B_r(p_0)$ but note there is non canonical choice for $p_0$ or $r$. We set $p_0$ to be the identity matrix $I_k$, $V = \frac{1}{k} I_k$, and $df = k$. Recall that since $\mathbb{P}(k)$ is negatively curved under the chosen metric, $r$ is finite but unconstrained. Operationally, however, there is no reason to believe that $\rho(y, I_k) \leq r$ for any chosen $r$, where the distance is manifold distance; we thus first set an $r$ and discard draws which are greater than distance $r$ from $p_0$ until we have sufficient draws for a desired sample size. For the simulations we set $k = 2$ and $r = 1.5$.

### 2.3.2 Sampling from KNG on SPDM

To sample from KNG for mean estimation on SPDM we use Metropolis-Hastings, a Markov chain Monte Carlo method. Let vech$(\cdot)$ denote the vectorization of a symmetric matrix and vech$^{-1}(\cdot)$ denote its inverse. Recall that the dimension $k = 2$. At each iteration $i$ we generate a proposal $x'$ by first randomly drawing a matrix $v$ from the tangent space at the current stage $T_{x_i}\mathbb{P}(2) \cong Sym_2$, and moving along $\mathbb{P}(2)$ in the direction of $v$ using the exponential map by proposing $f(x'|x_i) = \exp(x_n, t\sigma v)$ where $t \in (0, 1]$. We sample $v$ as $v = \text{vech}^{-1}(\tilde{v})$, where $\tilde{v}$ is $k(k+1)/2 = 3$-dimensional vector of uniform random variables on $[-0.5, 0.5]$; this indeed is not the same as a uniform draw on $Sym_2$. We then accept or reject the proposals of $f$ producing a Markov chain for the density $g(x) \propto \exp\{-\|U(x; D)\|_x/\sigma\}$. Here $\sigma = 2\Delta/\epsilon = 4r/n$ since $\epsilon = 1$ and $\Delta$ is as in Theorem ??.

1. Initialize $x_0 = \bar{x}$.

2. At the $i$th iteration, draw a matrix $v \in Sym_k$, the tangent space of $x_i$, as described above.

3. Generate a proposal $x'$ by letting $x' = \exp(x_i, t\sigma v)$.

4. Accept $x'$ and set $x_{i+1} = x'$ with probability $g(x')/g(x_i)$. Otherwise, reject $x'$ and generate a new candidate by returning to the previous generation step.

5. Return to step 2 until a chain of sufficient length has been created.

For our simulations we tuned $t$ at each sample size, but had a minimal 5000 burn-in steps and a thinning jump width of approximately 5000 to avoid correlation between adjacent accepted samples.

### 2.3.3 Choosing the ambient space radius

We compare privatization of our mechanism over $\mathbb{P}(k)$ to privatization in the ambient space of symmetric matrices $Sym_k$. To do this, we need to compute a comparable sensitivity. That is, given our data $D \subset B_r(I_k) \subset \mathbb{P}(k)$ we need to find $r_E$ of $D \subset \mathcal{B}_{r_E}(I_k) \subset Sym_k$, the radius of the geodesic ball in the space of symmetric matrices. Given the ball is centered at the identity matrix, it turns out that $r_E = e^r - 1$ as shown in [2].

## 2.4 Sphere simulations

### 2.4.1 Generating random samples

To generate random samples in $B_r(p_0) \subset \mathcal{S}_1^2$ we use polar coordinates. First, let $(\theta, \phi)$ be the pair of angles where $\theta \in [0, \pi]$ is the radial coordinate and $\phi \in [0, 2\pi)$ is the polar angle. We uniformly sample on $\theta \in [0, r]$ and $\phi \in [0, 2\pi)$ and set $r = \pi/8$. This results in data in $B_r(p_0)$ where $p_0$ is the north pole and higher concentration of data nearer $p_0$.

### 2.4.2 Sampling from KNG on $\mathcal{S}_1^2$

To sample from KNG on $\mathcal{S}_1^2$ we use a Metropolis-Hastings algorithm in a manner similar to that described above in Section 2.3.2 with the only difference being how we make proposals. At each iteration $i$ we generate a proposal $x'$ in the following manner. First, we draw a sample direction by drawing a vector from $N_3(\mathbf{0}_3, \mathbb{I}_3)$, scale this vector to have length $\sigma$, and then project this vector onto the tangent space $T_{x_i}\mathcal{S}_1^2$ of $x_i$ to produce a vector $\tilde{v}$. We then make a proposal by setting $x' = \exp(x_i, t\tilde{v})$ with $t \in (0, 1]$. The projection onto the tangent space ensures that $\|\tilde{v}\| \leq \sigma$.

Here $\sigma = \Delta/\epsilon$ where $\Delta = 2r(2 - h(r, \kappa))/n$ and $\epsilon = 1$. For our simulations we set $t = 0.5$, a burn-in period of 20 000 and thinned the chain every 600 to avoid correlated adjacent samples.

## 2.5 Kendall's 2D shape space simulations

### 2.5.1 KNG over Kendall's shape space

To sample from KNG on the space of Kendall shape space we use a Metropolis-Hastings algorithm similar to that described in Section 2.3.2. Let $k$ be the number of landmarks for our set of shapes. At each iteration $i$ we generate a proposal $x'$ as follows:

1. Sample $v = \{v_i\}$ such that the real and imaginary components of each $v_i$ are independent draws from $U(0,1)$.

2. Set $\tilde{v} = v - \frac{1}{k}\sum v_i$, which is simply $v$ centered at the origin.

3. Compute the horizontal component of $\tilde{v}$ on the tangent space of $x_i$, denote this as $\tilde{v}_h$.

4. Propose $x' = \exp(x_i, t\tilde{v}_h)$.

For our experiments we have a burn-in period of 7500.

We make two key assumptions that are needed for computing the sensitivity. First, we set $\kappa_{\max} = 4$ which is the maximal curvature of Kendall shape space. This perhaps can be improved upon, but we only require an upper bound on the curvature as per Assumption **??**, so we assume a worst case scenario. We set $r = \max_i \rho(\bar{x}, x_i)$ the maximum shape distance from the Fréchet mean shape and all landmark configurations in our dataset.

### 2.5.2 Shape point-wise Laplace

Suppose we have a dataset $D = \{x_1, x_2, \ldots, x_n\}$ such that $x_i = \{x_{i,j}\} \in \mathbb{C}^k$ is an set of $k$ labelled landmarks. We assume the shapes are all centered and scaled as in **??**. To compute a sanitized estimate in Euclidean space we sanitize each coordinate in the following manner.

1. Compute the Fréchet mean in shape space, denote this as $\bar{x} = \{\bar{x}_j\}$

2. Rotationally align each shape $x_i$ to the mean $\bar{x}$ using Procrustes analysis, denote this as $\tilde{x}_i$. That is, $x_i \to Ox_i$ where $O = \mathrm{argmin}_{O \in SO(2)} \rho(\bar{x}, Ox_i)$,

3. For each landmark $j$, find the maximal distance from $\bar{x}_j$ in the real and imaginary direction, call these $d_{x,j}$ and $d_{y,j}$.

4. Sanitize each landmark of the mean in the real and imaginary direction using the standard Laplace.

Similar to Section 2.5.1 above, rather than assume an $r$ for the ball in which the data lives, we determine this by setting this as a maximum distance to each landmark. So, at each landmark and in each direction we have $\sigma = \Delta/(\epsilon/2k) = 4rk/n$ where we set $r = d_{x,j}$ for the real direction and $r = d_{y,j}$ in the imaginary direction. We divide the privacy budget by $2k$ since we sanitize each landmark in each coordinate. By dividing the privacy budget among the landmarks in this way, the entire shape will have total privacy budget $\epsilon$. We compute the orthogonal alignment using standard Procrustes analysis.