# OpenReview forum: "Shape And Structure Preserving Differential Privacy"
_NeurIPS.cc/2022/Conference — NeurIPS 2022 Accept_

### Official Review · Reviewer_sqTu · 2022-07-04

**Rating:** 7
**Confidence:** 3
**Soundness:** 3 good
**Presentation:** 2 fair
**Contribution:** 3 good

**Summary:**

The authors of this paper study the problem of computing Frechet mean on manifold with differentially privacy constraints. The problem is interesting and the results are promising for other related problems.



**Questions:**

It will be good to compare with an iterative based algorithm for manifold optimization (i.e., when you compute the Frechet mean), at least in the numerical section. See a related paper:

Han, Andi, et al. "Differentially private Riemannian optimization." arXiv preprint arXiv:2205.09494 (2022).


**Limitations:**

I suggest the authors to spend more effort on making the paper more clear to the first readers (especially Section 3). The reviewer believes that this paper can attract the community if written nicely.

**Strengths And Weaknesses:**

Geometrical interpretation will be very helpful especially for the first readers.


Section 2.2, I think the description of the first equation is not correct.

---

> ### Author Response · Authors · 2022-08-02
> **We thank the reviewer for their comments.**
>
> Comment:Section 2.2, I think the description of the first equation is not correct.
>
> Reply: We aren't sure what the reviewer is referring to here, but if more detail is provided then we can provide a response.  The first equation states that many statistical/ML problems can be viewed as maximizing a utility function (or equivalently, minimizing a loss function).  We think this is a reasonable description.
>
>
> Comment: It will be good to compare with an iterative based algorithm for manifold optimization (i.e., when you compute the Frechet mean), at least in the numerical section. See a related paper: Han, Andi, et al. "Differentially private Riemannian optimization." arXiv preprint arXiv:2205.09494 (2022).
>
> Reply: We agree this would be an interesting direction and appreciate the reviewer pointing out that work. However, we have to stress that the arXiv version of that paper was posted after the Neurips deadline and thus we did not have the chance to consider it in our work.  Regardless, we will add the reference to our paper to help flesh out the importance of manifold methods in statistics/ML.

---

### Official Review · Reviewer_EM5Z · 2022-07-10

**Rating:** 6
**Confidence:** 3
**Soundness:** 3 good
**Presentation:** 2 fair
**Contribution:** 2 fair

**Summary:**

This work proposes a privacy mechanism in the Riemannian manifold, where the output decided by the dataset is determined by the Fréchet mean estimation. The authors focus on the structure of the utility function and derive the sensitivity in the manifold. A KNG privacy mechanism is used for the problem and some theoretical analysis based on the KNG and manifold is provided in the paper. The examples of spheres, SPD matrix, and Kendall’s 2D shape space are proposed. The application in Kendall’s space is interesting and helpful in reality. It also gives us some insights to understand the practicality of the proposed method.

**Questions:**

Some questions:
1. How is the sensitivity analysis combined with privacy analysis? Or just providing a global sensitivity for existing methods.
2. Can more detailed and local sensitivity analysis help a tight privacy analysis for KNG?  The sensitivity is global for all $x$ in the analysis.
3. How large $c_1$ and $c_2$ are in Theorem 3? Are they related to other constants? Some estimation or examples should be provided for some insight of the magnitude of $c_1$ and $c_2$.


Some suggestions:
1. The technical challenges can be clarified in the main context, especially compared with KNG and  Laplace mechanism in [Reimherr et al., 2021]).
2. More detailed preliminaries should be given for Kendall’s experiments.
3. The experiments for the verification of the proposed magnitude can be conducted.


**Ethics Review Area:**

["I don’t know"]

**Limitations:**

The authors show some limitations of their work with a clarification of the assumptions.

**Strengths And Weaknesses:**

Pros:
1. The proposed methods can solve the privacy problem in Fréchet mean estimation and reveal the influence of the curvature, which can be formulated as many problems in the real world.
2. The methodology and the manifold structure/sensitivity are well studied theoretically in the work, which improves the reliability for the usage.

Cons:
1. The technical challenges of the work are not clear. The novelty in techniques needs to be restated to show that the methodology/analysis is not a simple combination of existing methods (such as differences between KNG or Laplace mechanism in [Reimherr et al., 2021]).
2. Some parts especially in Kendall’s experiments should be reorganized. Some preliminaries should be stated more clearly. It will be more friendly to some readers.

---

> ### Author Response · Authors · 2022-08-02
> **We will make sure that the technical challenges are clear as well as the differences between our work and that of Reimherr et all 2021.**
>
> Comment: The technical challenges of the work are not clear. The novelty in techniques needs to be restated to show that the methodology/analysis is not a simple combination of existing methods (such as differences between KNG or Laplace mechanism in [Reimherr et al., 2021]).
>
> Reply:
> We can make the technical challenges more apparent in the paper. Among them sampling from KNG was particularly difficult in the space of SPDM partially due to needing to take the (-1/2) power of matrices at each step of the MCMC. For spheres and other positively curved manifolds such as Kendalls shape space, the curvature was shown to pose an issue in Reimherr et al. (2021), KNG however partially bypasses the issue by use of the gradient of the energy function giving better utility guarantees both theoretically and as we see numerically.
>
> KNG may seem like a combination of the Laplace however that is somewhat incidental. Rather than working with a distance in the exponent as in the Laplace, or a utility function in the exponent as in the exponential mechanism, we use the gradient of a general utility function. In this paper we only focused on mean estimation to do a direct comparison with Reimherr et al. (2021) however we can use a utility function for other summary statistics.
>
>
> Questions
> 1. How is the sensitivity analysis combined with privacy analysis? Or just providing a global sensitivity for existing methods.
>
> The two are very much interconnected and our methods are not just a global sensitivity for existing methods.  We provide a new methodology and to prove that this methodology is DP we require a very specific sensitivity analysis.  We will make this more clear in a revision.
>
> 2. Can more detailed and local sensitivity analysis help a tight privacy analysis for KNG? The sensitivity is global for all $x$ in the analysis.
>
> We have focused on global sensitivity.  We have yet to consider local/smoothed sensitivity however this would be an interesting direction for further research.  For more complicated manifolds with non-constant curvatures there could be some interesting opportunities to look at local sensitivities.  We will add this note to our discussion.
>
> 3. How large are $c_1$ and $c_2$ in Theorem 3? Are they related to other constants? Some estimation or examples should be provided for some insight of their magnitude.
>
> Constants $c_1$ and $c_2$ are upper and lower bounds on the determinant of the matrix representation of $\text{d}\exp(\bar x,\cdot)$, the differential of the exponential map at the mean $\bar x$. It is not possible to comment on their magnitude except for specific manifolds. For example, when the manifold $\mathcal M$ is a Lie group, $c_1$ and $c_2$ directly depend on magntiude of the eigen values of $\frac{\mathbb{I} - \exp(-ad_{\bar x})}{\text{ad}_{\bar{x} }}$, where
>
> $\text{ad}_{\bar{x}}$ is the adjoint representation at $\bar{x}$.
>
> In general, our goal was only to demonstrate that one obtains the desired asymptotic rates of convergence.  A more careful finite sample analysis would be prudent if one wanted a better handle on $c_1,c_2$ and any other constants.  Note also that asymptotically the loss function concentrates around a single point and thus $c_1$ and $c_2$ depend on the sample size (and converge to 1).

---

### Official Review · Reviewer_Z7Fo · 2022-07-12

**Rating:** 6
**Confidence:** 3
**Soundness:** 3 good
**Presentation:** 3 good
**Contribution:** 2 fair

**Summary:**

The paper provides a mechanism to achieve differential privacy of objects on a Riemannian manifold. By leveraging the connection between Riemannian distance and the exponential map on a manifold, the paper presents a novel generalization of the K Norm gradient mechanism for attaining DP extended to the scenario of Riemannian manifolds. The authors demonstrate an example of the utility of the low-rank preserving structure of their approach through a real-data experiment in shape-spaces by showing that a generic structure-unaware DP mechanism may be unable to capture the essential features of the objects in the dataset, while the methods developed in this paper succeed in doing so. The mechanism also allows for a potential computational gain in sampling because of its low-rank structure preserving nature.

**Questions:**

1. Can you provide more example areas of applications in terms of specific problems that can be addressed by this mechanism. Even though one such example is mentioned, I feel extending on the list of application areas can improve the visibility of the work beyond shape analysis.

2. Can you give an intuition what issues non-existence or non-uniqueness of Frechet means may create? How would it affect the sensitivity of the mechanism? I am curious as to what could be an application example of such a case.



**Limitations:**

The authors have not addressed limitations and potential negative impacts of their work. However, I feel this is an important area and we should be mindful of any negative outcome.  Unfortunately, I am not an expert in the area and I would leave the precise explication of the limitations to experts.

**Strengths And Weaknesses:**

In general, I feel the paper is well-motivated and its contribution is clearly-explained, the contributions are novel and have a wide range of applications for usage in various problems. The authors provide an example of one such application area, namely, that of shape-preserving transformations. Overall, I feel,  the authors have done a good job in clearly explaining the details of the paper in simple terms.

However, I feel there needs to be more examples of applications to motivate and broaden the scope of this work.

---

> ### Author Response · Authors · 2022-08-02
> **We will add more examples of commonly encountered manifold data such as statistical shape analysis and graphs.**
>
> Questions:
>
> 1. Can you provide more example areas of applications in terms of specific problems that can be addressed by this mechanism. Even though one such example is mentioned, I feel extending on the list of application areas can improve the visibility of the work beyond shape analysis.
>
> Yes and we can add this to the paper.  In brief, manifold-valued data arises is various applications these days. Our shape example came from a biomedical application (for more on this, see the book \emph{Statistical Shape Analysis} by I. L. Dryden and K.V Mardia). In computer vision, data arises as observations from the rotation and affine groups, both differentiable manifolds (e.g. pose estimation, synchronisation problems; a good reference for this is the book \emph{Numerical Geometry of Images: Theory, Algorithms, and Applications} by R. Kimmel (2003); in network and graph analysis, statistics on samples of networks is carried out by representing networks with their Laplacian matrices, which constitute a subset of the (stratified) manifold of positive semidefinite matrices (see for e.g., Ginestet, C. E et al. (2017). \emph{Hypothesis testing for network data in functional neuroimaging}. Ann. Appl. Stat. 11 725–750). Our proposed k-norm mechanism can be used to implement differential privacy  (with respect to the Frechet mean) in such settings.
>
> Further, we only consider a utility function for estimation of the mean however the mechanism can use any general utility function. Other statistical problems which can be posed as solutions to a utility function include linear regression and median estimation. We can mention these in the paper as well.
>
>
> 2. Can you give an intuition what issues non-existence or non-uniqueness of Frechet means may create? How would it affect the sensitivity of the mechanism? I am curious as to what could be an application example of such a case.
>
> As a simple example, the uniform distribution over a sphere has no unique mean as all points would have the same energy; a sample from the uniform distribution on a sphere likely will have at least one sample mean however the sensitivity is effectively unbounded as a single point can move the mean to an antipodal position.

---

### Official Review · Reviewer_gDXf · 2022-07-19

**Rating:** 5
**Confidence:** 5
**Soundness:** 4 excellent
**Presentation:** 4 excellent
**Contribution:** 2 fair

**Summary:**

This work studies a class of statistical estimation problems of sensitive data $D$ on Riemannian manifolds $\mathcal{M}$, which can be formulated as minimizing a utility function $U(x; D)$ over $x\in\mathcal{M}$. To this end, the authors apply the K-norm Gradient Mechanism (KNG), proposed by Reimherr and Awan (2019), which applies the well-known exponential mechanism to the gradient norm $\lVert \nabla U(x; D) \rVert_x$.

The authors focus on the problem of privately estimating the Fréchet mean, in which case the utility function that we want to minimize is the variance functional $F(x; D)$. With the KNG mechanism and an assumption that $D$ is contained in a geodesic ball with radius bounded by 1) the injectivity radius and 2) the inverse of the sectional curvature of $\mathcal{M}$, they derive the sensitivity of the norm of gradient vector field $\lVert \nabla F(x; D) \rVert_x$. Here, the sensitivity $\Delta$ is dependent in the sectional curvature of $\mathcal{M}$. This results in a differentially private algorithm for estimating the Fréchet mean via sampling from the density $f (x; D) \propto \exp(-\epsilon\lVert \nabla F(x; D) \rVert_x/2\Delta )$. Assuming further that there is a lower bound for the sectional curvature of $\mathcal{M}$, the authors show that the utility of this mechanism is $O\left(\frac{d}{n \epsilon}\right)$ which matches the lower bound in the Euclidean case.

The authors also perform several experiments:
1. A simulation on the $d$-dimensional sphere, which has positive sectional curvature
2. A simulation on the space of $k\times k$ symmetric positive-definite matrices with the Rao-Fisher metrix, which has negative sectional curvature
3. 2D Statistical shape space analysis on the data of mid-sagittal slices of MRIs,

which demonstrate that the proposed method can be better utility compared to the previous DP algorithms on Riemannian manifolds.



**Questions:**

In Assumption 1, when the manifold has negative sectional curvature, it states that $r$ is unbounded. But we still have $r< \frac{1}{2} \text{inj }\mathcal{M}$, right? So $r$ must not be unbounded, unless I am missing some fact about negatively curved manifolds.

**Limitations:**

The authors have addressed the limitation of the KNG mechanism in the Conclusions, including the difficulties of sampling from the KNG.

The data of MRIs is a public dataset, so there is no ethical concerns here.

**Strengths And Weaknesses:**

# Strenghts

This paper provides a nice way of extending the differential privacy analysis of the exponential mechanism from the Euclidean spaces to Riemannian manifolds. It is interesting to see that using $-\lVert \nabla F(x; D) \rVert_x$ as the score function in the exponential mechanism is better than using $- F(x; D)$ directly. I find the writing to be very clear and concise, given the complex subjects. I appreciate the authors' effort to include manifolds with both positive and negative sectional curvatures in the experiments, as well as an application on the real data.

Small note: for any compact Riemannian manifold, the sectional curvature is bounded (and hence lower bounded), so it satisfies the assumption in Lemma 1.

# Weaknesses

This work seems to be quite incremental, as the proof of sensitivity bound (Theorem 2) is already given in the supplemental to Reimherr et al (2021). So the main contribution of this paper lies in the utility analysis and the experiments, which is quite lacking for a DP paper in my opinion.

Another issues is the non-trivial sampling from the exponential mechanism: $f (x; D) \propto \exp(-\epsilon\lVert \nabla F(x; D) \rVert_x/2\Delta )$, which is done via Metropolis-Hastings (MH) in this paper. However, the sampling distribution of the MH is not exactly the same as that of the exponential mechanism, so there is no guarantee that MH satisfies $\epsilon$-DP. Right now, the best we can do is $(\epsilon,\delta)$-DP, where $\delta$ depends on the total variation between the MH and target distribution (see Shen & Yu, 2013, Lemma 2 and Wang et al., 2015, Proposition 3). With careful analysis, it is possible to obtain $\epsilon$-DP in the case of Gibbs sampling (Foulds et al., 2016, Section 4.1).

The fact that sampling from $f (x; D)$ is non-trivial, and how one can resolve this issue, could have been mentioned in the earlier sections.

References:
* Foulds, J., Geumlek, J., Welling, M. and Chaudhuri, K. 2016. On the Theory and practice of privacy-preserving Bayesian data analysis. UAI'16.
* Shen, E. and Yu, T. 2013. Mining frequent graph patterns with differential privacy. KDD ’13. doi: 10.1145/2487575.2487601.
* Wang, Y.-X., Fienberg, S. E., and Smola, A. J. 2015. Privacy for free: Posterior sampling and stochastic gradient monte carlo. ICML’15.

---

> ### Author Response · Authors · 2022-08-02
> **Rather than incremental we see this as a key improvement for the case of positively curved manifolds as compared to Reimherr et al 2021. Our utility guarantees for positively curved manifolds and the adaptability of the mechanism to a gradient of a general loss function is fundamental to manifolds commonly encountered in statistics and ML.**
>
> Comment: This work seems to be quite incremental, as the proof of sensitivity bound (Theorem 2) is already given in the supplemental to Reimherr et al. (2021). So the main contribution of this paper lies in the utility analysis and the experiments, which is quite lacking for a DP paper in my opinion.
>
> Reply:
> A key motivation for this paper lies in the need to obtain better utility on positively curved manifolds when compared to that obtained by Reimherr et al. (2021). In the final paragraph of their paper they specifically point this out as an avenue for future work.
>
> Within this context, we have demonstrated, theoretically and numerically, that it is possible to exercise better control over utility calculations by replacing $F$ with its gradient $\nabla F$. In other words, the fact that proof of Theorem 2 follows easily from Reimherr et al. (2021), while true, is not the main theoretical contribution; it is that we are able to obtain a $\textit{better}$ upper bound by getting rid of the extra factor $h_{\max}$ that affected the bound in Reimherr et al. (2021) (Remark 1), and then demonstrating through Lemma 1 and Theorem 3, the additional gain in utility (Figure 1).
>
> Concerning utility analysis and experiments, we note that we have considered as examples two positively curved manifolds that commonly arise in data analysis: spheres and the 2D Kendall landmark shape space, identified with a suitable complex projective space. To our knowledge, the only other commonly encountered (in Statistics and Machine Learning) positively curved manifolds we have not examined are the rotation group and the Stiefel manifold, both of which are `similar' to spheres and complex projective space (they too are Einstein manifolds with positive curvature everywhere); we thus expect similar gains in utility, and this can be verified numerically.
>
> Our paper thus significantly adds to the nascent literature on differential privacy on manifolds, by not only showing how a k-norm mechanism can be developed, but more importantly, demonstrating that for the particularly interesting case of positively curved manifolds when the nonlinear manifold structure plays a key role, it can be a better choice when compared to the current state-of-the-art (Laplace mechanism by Reimherr et al. (2021)).
>
> Yes, you are right about bounded sectional curvatures for compact manifolds--thanks for pointing it out. We will add a note on this in the paper, and remark that the bounds will influence the sensitivity.
>
> As for the sampling via the MH hastings algorithm, we agree it is worth mentioning and referencing the difficulty in sampling earlier in the paper. Classical results, e.g. Mengersen and Tweedie (1996), suggest that MH is, under some mild regularity conditions, geometrically ergodic, thus the incurred $\delta$ should be practically very small given the lengths of our chains (though this is still an active area of research).  We will expand upon this discussion in a revision.
>
>
>
> Question: In Assumption 1, when the manifold has negative sectional curvature, it states that $r$ is unbounded.
>
> Reply:
> We do not mean that $r$ can be unbounded and we will clarify this point. It can happen that for negatively curved manifolds $\text{inj}\mathcal M=\infty$, and thus the manifold structure places no additional restrictions on $r$. However, a finite $r$ is still needed to bound the global sensitivity just as in the Euclidean case.

---

> > ### Comment · Reviewer_gDXf · 2022-08-09
> > **Reply**
> >
> > Thanks. This clears up my concern on the main contribution of the paper.
> >
> > However, there is still one remaining concern: there is a gap between the theory and practice, as sampling with MH algorithm might not give us the privacy guarantee of the exponential mechanism (EM). How large is the constant in the convergence analysis in Mengersen and Tweedie (1996)? If it is large, then how long does it take for $\delta$ to be small?
> >
> > Note that this gap is not only specific to this paper, but also a major concern to researchers who want to employ EM for DP optimization algorithms. Here is an excerpt from [1]:
> >
> > > ...one can construct simple score functions on the hypercube for which the natural Metropolis chain run for any polynomial time leads to a non-private algorithm [Ganesh and Talwar, 2019]. There are also well-known complexity-theoretic barriers in exactly sampling from exp(−f ) if f is not required to be convex.
> >
> > Thus we should be very careful about approximate sampling of EM, especially when working on manifolds. In this regard, there have been some recent works that try to resolve this issue [1,2,3].
> >
> > I think that this paper is one step away from being a really great paper; as of now, there is no way to guarantee that the approximate sampling can be private.
> >
> > * [1] Ganesh, A., & Talwar, K. (2020). Faster differentially private samplers via Rényi divergence analysis of discretized Langevin MCMC. Advances in Neural Information Processing Systems, 33, 7222-7233.
> > * [2] Seeman, J., Reimherr, M., & Slavković, A. (2021). Exact Privacy Guarantees for Markov Chain Implementations of the Exponential Mechanism with Artificial Atoms. Advances in Neural Information Processing Systems, 34, 13125-13136.
> > * [3] Gopi, S., Lee, Y. T., & Liu, D. (2022). Private convex optimization via exponential mechanism. arXiv preprint arXiv:2203.00263.

---

### Meta-Review · Area_Chair_Uoti · 2022-08-21

**Recommendation:** Accept
**Confidence:** Less certain

**Metareview:**

The reviewers agree that the paper should be accepted (albeit with a mix of the level of acceptance). I agree. The presentation of the paper could be better.

**Award:**

No

---

### Decision · Program_Chairs · 2022-09-14

Accept